# Nutritional Ergogenic Aids in Combat Sports: A Systematic Review and Meta-Analysis

**DOI:** 10.3390/nu14132588

**Published:** 2022-06-22

**Authors:** Néstor Vicente-Salar, Encarna Fuster-Muñoz, Alejandro Martínez-Rodríguez

**Affiliations:** 1Biochemistry and Cell Therapy Unit, Institute of Bioengineering, University Miguel Hernandez, 03201 Elche, Spain; 2Department of Applied Biology-Nutrition, Miguel Hernandez University, Alicante Institute for Health and Biomedical Research (ISABIAL), 03203 Alicante, Spain; e.fusterumh@gmail.com; 3Department of Analytical Chemistry, Nutrition and Food Science, University of Alicante, Alicante Institute for Health and Biomedical Research (ISABIAL), 03690 Alicante, Spain

**Keywords:** combat sports, ergogenic aid, performance, sport supplement

## Abstract

Nutritional ergogenic aids (NEAs) are substances included within the group of sports supplements. Although they are widely consumed by athletes, evidence-based analysis is required to support training outcomes or competitive performance in specific disciplines. Combat sports have a predominant use of anaerobic metabolism as a source of energy, reaching peak exertion or sustained effort for very short periods of time. In this context, the use of certain NEAs could help athletes to improve their performance in those specific combat skills (i.e., the number of attacks, throws and hits; jump height; and grip strength, among others) as well as in general physical aspects (time to exhaustion [TTE], power, fatigue perception, heart rate, use of anaerobic metabolism, etc.). Medline/PubMed, Scopus and EBSCO were searched from their inception to May 2022 for randomised controlled trials (RCTs). Out of 677 articles found, 55 met the predefined inclusion criteria. Among all the studied NEAs, caffeine (5–10 mg/kg) showed strong evidence for its use in combat sports to enhance the use of glycolytic pathways for energy production during high-intensity actions due to a greater production of and tolerance to blood lactate levels. In this regard, abilities including the number of attacks, reaction time, handgrip strength, power and TTE, among others, were improved. Buffering supplements such as sodium bicarbonate, sodium citrate and beta-alanine may have a promising role in high and intermittent exertion during combat, but more studies are needed in grappling combat sports to confirm their efficacy during sustained isometric exertion. Other NEAs, including creatine, beetroot juice or glycerol, need further investigation to strengthen the evidence for performance enhancement in combat sports. Caffeine is the only NEA that has shown strong evidence for performance enhancement in combat sports.

## 1. Introduction

The different disciplines of combat sports have as common elements the involvement of explosive and high-intensity movements of both the upper and lower limbs. They are executed in rounds or bouts of a short duration (seconds to minutes), the objective of which will be conditioned by the specific combat sport and its rules [1]. Performance in combat sports is determined by the acquisition of a physical and physiological profile appropriate to the characteristics of each discipline [2]. The requirements of combat sports involve a great technical record and a high number of repetitions at high intensity, which are interspersed with moments of low intensity. The energetic demands of these sports show a high involvement of aerobic metabolism. However, high-intensity actions require the involvement of anaerobic metabolism (the ability to produce energy by intramuscular adenosine triphosphate (ATP) and phosphocreatine and/or anaerobic glycolysis during short duration exercise) [1], represented by elevated blood lactate levels after competition [2].

The magnitudes of strength involved differ among combat sports. On the one hand, those with predominantly punching movements, such as punches and kicks, have a greater need for explosive strength and power; on the other hand, those with grappling actions may require a greater emphasis on isometric and concentric muscle strength. Likewise, there are also differences according to the limbs that mainly develop the motor actions: in sports such as boxing or judo, the upper limbs are mostly used; in taekwondo it is centred on the lower limbs, while in karate both upper and lower limbs are used. In each of these sports, specific equipment and clothing are used that, according to scientific literature, could condition the technical aspects and even the physical demands of the athletes [2]. Depending on these characteristics, which are based, among others, on the rules of the sport, combat sports can be categorised into two main groups: grappling and striking. In addition, a third group involves both grappling and striking, namely mixed martial arts (MMA) [2].

Within the combat sports with grappling, there are sports in which gripping, throwing, ground combat, chokeholds and joint locks are allowed. These sports include judo, wrestling and jiu-jitsu [3,4]. Punching or striking sports include those in which the hands (such as boxing), the legs (such as taekwondo) or both the hands and legs (such as karate) are used [2]. On some occasions, namely non-Olympic modalities, knee and elbow strikes are involved [5]. MMA allow both grappling and striking techniques following the specific rules of this sport. Similar to the previously mentioned sports, aerobic and anaerobic metabolism are involved in the development of this discipline, due to the repetition of high-intensity efforts and combining the demands of grappling and striking [6]. It is important to highlight that competitions in these disciplines are by weight category, which are established to match combat sports athletes with similar physical characteristics and thereby emphasise fair play. Most of the athletes try to reach the maximum (limit) weight of the category. In addition, inappropriate weight loss strategies may negatively affect performance [7].

An ergogenic aid is defined as a psychological technique, mechanical device, nutritional or pharmacological approach that can improve training adaptations and/or exercise performance capacity [8]. Nutritional ergogenic aids (NEAs) are taken orally; they contain nutritional ingredients whose objectives are to improve sports performance and to avoid harmful effects on the individual (i.e., extraneous fatigue, dehydration and loss of physical skills, among others) [9]. The consumption of NEAs has increased in recent years around the world, despite a 32.1% decline in sports supplements sales in 2020, the first year of the COVID-19 pandemic. It has been estimated that sports supplements sales will increase by 10–11% between 2022 and 2028 [10]. There are no differences in supplement consumption by sex, but elite athletes usually take more dietary supplements than non-elite athletes [11]. There is no specific information about the consumption of supplements among combat sports practitioners, but there is no doubt about the potential ergogenic effect of these dietary supplements in specific sport modalities, because it is conditioned by the type of effort executed [12,13]. However, proper counselling based on current scientific evidence is required.

Several organisations such as the World Anti-Doping Agency (WADA) and the Australian Institute of Sport (AIS) have proposed classifications of sports supplements, grouping them into different categories according to safety, legality and effectiveness. Nevertheless, there are no policies regarding the regulation of alleged benefits and safety claims, so in many cases, companies advertise their products without scientific evidence regarding their effect, dose or instructions for use [14,15]. This systematic review and meta-analyses aimed at evaluating the scientific evidence concerning NEAs in the improvement of the performance of combat sports athletes specifically through published randomised controlled trials (RCTs).

## 2. Materials and Methods

The conduct and reporting of the current systematic review conforms to the Preferred Reporting Items for Systematic Reviews and Meta-Analyses (PRISMA) [16]. Using MeSH terms, three groups of combat sports were analysed regarding the efficacy of certain NEAs: martial arts, boxing and wrestling.

### 2.1. Systematic Search

The electronic databases Medline, Scopus and EBSCO (Sportdiscus) were searched to find relevant articles identified by title and abstract (from inception to 31 May 2022) using the search strategy described in Table 1. To perform a complete search, reference lists from reviews and relevant publications were manually searched to find additional publications on the subject.

### 2.2. Data Extraction

Two reviewers (N.V.S. and E.F.M.) independently extracted the characteristics of the retrieved RCTs and the outcomes of interest from full-text articles. A third author (A.M.R.) assessed inter-reviewer differences (i.e., in the case of selecting an article in which the authors evaluated a multi-ingredient supplement to evaluate its synergy but it could have masked the real effect of the NEA). The following data were extracted using a predefined Microsoft Excel data extraction form: type of NEA, dose and time, the number of participants within each group, participant age and sex, combat sport discipline, measurement methods and main outcomes. This endeavour provided an overview table of all eligible studies.

### 2.3. Study Selection

The inclusion criteria were: (a) no use of doping substances established by the WADA; (b) an RCT design that included one group receiving NEAs and ≥1 group(s) receiving a placebo or not receiving supplementation; (c) not including any ergogenic aids classified within group A by the AIS because of their high evidence grade [17]; (d) not taking supplementation as a source of nutrients, such as bars, gels or drinks rich in carbohydrates, proteins and electrolytes; (e) not presenting medical supplementation to prevent or treat clinical issues; and (f) not grey literature (abstracts, conference proceedings or editorials) or reviews.

### 2.4. Quality Assessment and Publication Bias

Evaluation was carried out by two reviewers (N.V.S. and A.M.R.) working independently in order to assess risk of bias comprehensively. The characteristics of the retrieved RCTs were evaluated using the PEDro scale ‘risk-of-bias’ assessment tool [18]. The following bias criteria were considered: eligibility criteria, randomisation, allocation, baseline, blinding of subjects and researchers, measures of one key outcome from >85% of the allocated subjects, data from placebo and experimental groups and statistical comparisons, including variability of measurements. Disagreements were resolved by consensus involving a third reviewer (E.F.M.) following the recommendations of the Cochrane Handbook for Systematic Reviews of Interventions [19].

### 2.5. Statistical Analysis

A meta-analysis was performed to evaluate the effect of caffeine supplementation on blood lactate, using the Review Manager software (RevMan 5.3, Cochrane Collaboration, Oxford, UK). The authors used a Microsoft Excel template to collect the relevant information regarding the studies that included caffeine supplementation and blood lactate measurements. The template included the following variables: study authors and year of publication, sports discipline, group, caffeine dose and pre- and post-combat or test blood lactate data.

The effect of caffeine supplementation was determined by calculating the difference in the blood lactate before and after combat or a test. The blood lactate difference was subjected to inverse-variance weighting [20]. In addition, because not all the sports disciplines are the same, they do not use the same kind of combat. Hence, the standardised mean difference (SMD) was used and combined with a random effects model [21]. The 95% confidence interval (CI) was determined to evaluate the size of the changes.

I^2^ was calculated to evaluate the heterogeneity among the included studies. I^2^ < 25%, >50% and >75% were considered to indicate low, moderate and high heterogeneity, respectively. The Egger test was used to assess the publication bias by estimating the funnel plot asymmetry [22]. Statistical significance was set as *p* < 0.05.

## 3. Results

### 3.1. Included Studies

A total of 547 studies were screened by title and abstract and 439 were assessed for the eligibility criteria (full-text screening). From the retrieved articles, 55 met all inclusion criteria and were included in the systematic review (Figure 1). Of these 55 studies, 45 were found in Medline (two articles were not available despite requesting them from the corresponding author), seven were found in Scopus and three were retrieved from EBSCO (Sportdiscus) (Table 2). The PRISMA flowchart illustrating the step-by-step exclusion of unrelated/duplicate records, leading to the final selection of 55 RCTs that met the predefined inclusion criteria, is shown in Figure 1.

### 3.2. Risk of Bias and Quality Assessment of Studies

The total score of the PEDro scale was between 5 and 8 points out of 10, and the average was 7 points. Most of the included studies did not blind or did not report if assessors who measured at least one key outcome were blind to the treatments (Table 3).

### 3.3. Participants

The participants in the examined studies ranged from 16.1 to 29.9 years old, meaning that they included junior to senior athletes. The level ranged from amateur to professional in both sexes, with a majority of male athletes (*n* = 726) compared with female athletes (*n* = 78) and non-determined gender because the studies did not report it (*n* = 150) [27,34,37,45,57,58,63,64,65,69]. Most studies focussed on grappling disciplines (*n* = 33) [42,43,44,45,46,47,48,49,50,51,52,53,54,55,56,57,58,59,60,61,62,63,64,65,66,67,68,69,70,71,72,73,74], followed by striking disciplines (*n* = 15) [23,24,25,26,27,28,29,30,31,32,33,34,35,36,37,38,39,40,41], mixed disciplines (*n* = 1) [75] and combat sports in general since they were not identified in the methodology (*n* = 2) [76,77]. In the case of NEAs, caffeine was the most evaluated supplement (*n* = 26) [23,24,25,26,27,28,29,30,42,43,44,45,46,47,48,49,50,51,52,53,54,55,56,57,58], followed by plasma buffers (*n* = 19) [31,32,33,34,59,60,61,62,63,64,65,76], nitric oxide (NO) precursors (*n* = 8) [39,40,41,70,71,72,73,77], creatine (*n* = 4) [31,38,68,69] and hydration agents (*n* = 1) [74].

### 3.4. Nutritional Ergogenic Aids and Intervention Characteristics in Striking Combat Sports

Caffeine was the most tested NEA with eight studies (Table 2). All trials had a duration of 1 day with variations in concentrations and timing. Most studies used a caffeine dose of 5–6 mg/kg 30–60 min before the tests [24,26,27,28,29,30], with improvements in specific combat skills such as the effort–pause ratio and time of punching sequences during a simulated boxing combat and a greater number of attacks (+5–15% successful kicks or +27% in the number of attacks compared with the placebo group [25,30]) and less reaction and skipping time in simulated taekwondo combats compared with the placebo groups. General skills such as power (+2.3–6.8%), jump height (+5.1% [26]), time to exhaustion (TTE; +5.6% [27]), blood lactate (+20.0–26.2% [26,29,30]) and glycolytic energy contribution (+24.8–67.2% [29]) were higher than the placebo groups, while fatigue perception was lower (−61.5% [24]). Only one study used a lower caffeine dose (3 mg/kg 60 min before the test) and found an improvement in the number of kicks and the specific taekwondo agility test but without changes in fatigue perception [23]. Other protocols with mouth rinsing with 6 mg/kg of caffeine just before the tests in taekwondo athletes showed an increase in the percentage of successful kicks and less perceptual training intensity during a specific taekwondo test compared with the placebo group during a Ramadan period [25].

NEAs related to plasma buffer function have been evaluated by three 1-day and two 3–5-day studies. A load of 0.3 g/kg of sodium bicarbonate taken for 3 days before the test and intake of 0.1 g/kg 120, 90 and 30 min before a karate-specific test produced improvements in TTE (+8.9%), but maintenance of vertical jumps, blood lactate and heart rate compared with the placebo group [27]. In addition, for a higher sodium bicarbonate dose taken for 5 days (4 × 0.125 g/kg/day ≈ 0.5 g/kg/day), there were improvements in power (+18.4%) in a specific taekwondo test accompanied by a significant decrease in lactate after the test [31]. For the 1-day protocol, 0.3 g/kg of sodium bicarbonate taken 60–90 min before the test increased TTE (+55.5%), blood lactate (+39.5%), pH (+1.4%), glycolytic energy contribution, attack time and punch efficacy, with different results on the heart rate [32,33].

There has been only one study that combined caffeine with sodium bicarbonate regarding striking combat sports [27]. During a 3-day protocol, karate athletes ingested 0.3 g/kg/day of sodium bicarbonate. On the test day, they consumed 6 mg/kg of caffeine 60 min before and 0.1 g/kg of caffeine 120, 90 and 60 min before the test. The authors reported only a higher TTE (+9.3%) relative to the placebo group, while the highest vertical jump, blood lactate and heart rate were not different compared with the placebo group. Both NEAs consumed together did not show any advantage compared when they were consumed separately [27].

Beta-alanine, which has a role as an intracellular buffer precursor, has been used in three studies examining striking combat sports [35,36,37]. All the studies used boxers as volunteers. One study lasted 10 weeks with a dose of 4.9–5.4 g/day and showed improvements in general fitness (as peak power in lower limbs and less power drop in upper limbs, but not in strength or blood lactate levels) [36]. The other two studies, which lasted 4 weeks using 6 g/day or 0.3 g/kg/day of beta-alanine, observed mixed results in blood lactate levels compared with the placebo group [35,37]. Furthermore, the authors showed improvements using beta-alanine in specific combat skills, namely a higher number and force of punches, but only as a time × group interaction [37], and there were no changes in power or the fatigue index [35].

Two studies evaluated the efficacy of creatine monohydrate in taekwondo practitioners. A dose of 50 mg/kg/day for 6 weeks did not offer advantages compared with the placebo group regarding power or fatigue perception. It only produced a significant increase in blood triglycerides [38]. On the other hand, a higher dose over less time (4 × 5 g/day for 5 days) offered improvements in power (+17.3%) after a specific taekwondo test without affecting lactate levels [31].

Only one study has examined the combination of creatine and sodium bicarbonate, specifically in striking combat sports [31]. During a 5-day protocol, taekwondo athletes ingested 0.5 g/kg/day of sodium bicarbonate and 4 × 5 g/day of creatine. Compared with the placebo group, peak (+28.3%) and mean (+39.2%) power were significantly higher and blood lactate levels were lower after a specific test. Both NEAs consumed together showed advantages in mean power compared with when they were consumed separately [31].

There are studies that have examined NO precursors. Intake of 2 g/kg beetroot juice 150 min before the test in boxers did not produced differences in blood lactate levels or heart rate compared with the placebo group, but there was a significant decrease in power [41]. The same 1-day protocol with a controlled quantity of nitrate (NO_3_^−^; 400 or 800 mg) in taekwondo practitioners did not produce differences in the number of kicks, general fitness (TTE, jump height, anaerobic performance and heart rate), blood lactate levels or perceptual training intensity [39]. However, 1 g of beetroot extract improved VO_2_ peak (+10.0%) and the anaerobic threshold (+13.5%) compared with the placebo group without affecting blood lactate levels [40].

### 3.5. Nutritional Ergogenic Aids and Intervention Characteristics in Grappling Combat Sports

For striking combat sports, caffeine has been the most tested NEA in the grappling modality with 17 studies, with judo being the most evaluated discipline (Table 2). Caffeine has been tested in judoists 60 min before the exercise protocol with a dose ranging from 3 to 9 mg/kg [43,45,47,49,50,51,52,54,56,58]. Specific skills including the number of total throws [47,49,52,54,56], total attacks [49,50,51], the special fitness judo test (SFJT) index (expressed as heart rate divided by total throws) [52,56] and handgrip strength [43,45,51] showed mixed results regardless of the dose used. General skills or physiological responses such as heart rate [45,47,49,52,56], and fatigue or effort perception [45,47,49,50,52,54,56,58] also produced mixed results, while only blood lactate levels (+14.8–54.1% [47,51,54,56]) increased with a dose ranging from 5 to 6 mg/kg compared with the placebo groups [47,51,54,56]. Height or power of jumps did not change compared with the placebo group [51], but the reaction time and power showed significant improvements [58]. Velocity in the execution of some resistance exercises also was improved with the consumption of 3–6 mg/kg of caffeine [43]. Other grappling disciplines such as wrestling and Brazilian jiu-jitsu used similar caffeine protocols with doses ranging from 3 to 10 mg/kg 30–60 min before the tests [42,44,45,48,53,55,57]. In the case of combat skills, time to complete a specific wrestling test [48], the number of attacks but not defensive actions [55] and handgrip strength [45,55] were improved with 3–6 mg/kg of caffeine compared with the placebo group. Focussing on general physical aptitudes, wrestlers showed less improvement in jump height with a dose of 4–10 mg/kg of caffeine than Brazilian jiu-jitsu athletes, with better results in static lifts and weight, power and velocity in 1RM (increasing the repetitions in bench press compared with the placebo) taking 3 mg/kg of caffeine [42,44,48,53,55]. There were inconsistent changes in heart rate regardless of the caffeine dose [42,45,48,57], a trend to increase blood lactate for doss ranged from 3 to 6.2 mg/kg [48,55,57] and no changes in hydration status (based on urine osmolality and specific gravity) for doss < 10 mg/kg [48]. Lastly, in the case of effort perception, 5.2–6 mg/kg caffeine was optimal [48,53,55,57]. The use of caffeinated chewing gums (2.7–5.4 mg/kg of caffeine) by judoists 15 min before a test did not improve specific combat skills (total throws or the SFJT index) or general physiological aspects (blood lactate, heart rate or perceptual training intensity [46].

Different plasma buffers have also been evaluated, including sodium bicarbonate and sodium citrate [54,59,60,61,62,63,64,65]. A load of 0.5 g/kg of sodium bicarbonate offered for 7 days [62] or an increasing dose ranging from 0.025 to 0.1 g/kg g/kg provided for 10 days [60,61] produced a dose–response relationship in terms of improvements in power (+13.7–16.0% [62,63]) and total work (+8.0% [62]), but mixed results in terms of blood lactate levels. Even with a high dose there were no improvements in effort or fatigue perception compared with the placebo group [62]. Acute doses of sodium bicarbonate (0.1–0.3 g/kg) 60–120 min before tests improved specific skills such as throws (+4.9–5.1% [54,63]) in judoists in a dose–response manner [54,63] but not handgrip strength in jiu-jitsu athletes [59]. This dosing regimen produced higher power (+16.0% [63]) and significantly increased blood lactate levels (+17.8–26.3%) after specific judo tests [54,63] but not after anaerobic general tests (Wingate test) or handgrip and forearm strength contraction tests [59,63]. Similarly to long protocols with sodium bicarbonate, there were no changes in effort and fatigue perception with acute doses [54,63]. Sodium citrate has been evaluated in two studies using 0.6–0.9 g/kg 16 h, 8 h and 30–120 min before upper-body intermittent sprint performance tests in wrestlers and Brazilian jiu-jitsu practitioners [64,65]. Although the dose range elevated blood pH and improved water intake retention and plasma volume, only 0.9 g/kg of sodium citrate increased blood lactate levels (+15.0%) [64]. Power; heart rate; fatigue perception; and urine volume, osmolality and specific gravity were not significantly different compared with the placebo group.

Only one study used a combination of caffeine with sodium bicarbonate in judoists [54]. Participants ingested for 1 day 6 mg/kg of caffeine 60 min before the test with 0.1 g/kg of sodium bicarbonate 60, 90 and 120 min before a specific performance judo test. The number of throws was higher (+7.8% [54]) than the placebo group—and even compared with caffeine or sodium bicarbonate consumed separately—and the blood lactate levels were elevated (+21.9% [54]), while there were no significant differences in perceptual training intensity [54].

Beta-alanine, an intracellular buffer precursor, has been evaluated in three studies focussing on grappling combat sports [62,66,67]. A dose of 4.4 g/day for 8 weeks in amateur wrestlers did not improve general test performance (running time, flexed arm hang time or blood lactate levels) [67]. On the other hand, when the dose was increased to 6.4 g/day for 4 weeks, judoists and jiu-jitsu practitioners showed a higher number of throws (+9.0% [66]), higher total work (+7.0% [62]) and greater mean power (+6.5–10.5% [62]) (but not peak) but no changes in blood pH and perceptual effort [62,66].

To assess the possible synergistic effect with the combination of intramuscular and plasma buffers, judoists and jiu-jitsu volunteers followed a 4-week beta-alanine protocol using 6.4 mg/day with a 7-day sodium bicarbonate protocol (0.5 g/kg) [62]. In the general Wingate anaerobic test, the mean power (+8.6–20.3%) and peak power (+15.3–22.3%) were higher than the placebo group and during more bouts than sodium bicarbonate or beta-alanine consumed separately. Moreover, total work was even higher that the consumption of beta-alanine or sodium bicarbonate alone. Blood lactate was significantly higher than the placebo group and the perceptual training intensity was lower (taking these NEAs separately did not affect this parameter) [62].

The effect of creatine has been evaluated in two studies in amateur wrestlers and Brazilian jiu-jitsu practitioners, one of them with a load of 0.3 g/kg/day for 5 days [68], and the other for 15 days [69]. There were improvements relative to the control group in handgrip strength and agility [69] but no significant changes in peak and mean power, blood lactate levels, heart rate, fatigue perception or urine specific gravity [68].

Four studies have evaluated NO precursors: one of them using beetroot juice [70], another using a beetroot-based gel [71] and the other two examining different arginine protocols [72,73]. Ingestion of 600 mg of NO_3_^−^ contained in beetroot juice 150 min before an isokinetic strength test increased the peak of strength only in the upper limbs (+13.4–15.1%), while the mean strength was improved in the upper and lower limbs (+9.6–16.6%) [70]. Intake of 12.2 mmol of NO_3_^−^ contained in a beetroot-based gel by amateur Brazilian jiu-jitsu volunteers for 8 days resulted in a significant increase in maximal voluntary forearm contraction and muscle oxygen saturation during a forearm isometric test, and a significant decrease in blood lactate (−29.3%) after exercise [71]. Long protocols with 6 g/day of arginine for 3 days in judoists only increased blood arginine levels without affecting power and blood lactate after an intermittent anaerobic test [73]. In the case of acute protocols using 150 mg/kg of arginine 60 min before a cycloergometer test in wrestlers, there was only an improvement in TTE (+5.8% [72]) and no changes in blood lactate, heart rate or VO_2_ [72].

Lastly, among hydration agents only glycerol has been examined [74]. Consumption of 1 g/kg glycerol 60 min before the Wingate test by wrestlers did not alter anaerobic power, body mass, urine specific gravity and saliva osmolality.

### 3.6. Nutritional Ergogenic Aids and Intervention Characteristics in Other Combat Sports

Caffeine as an NEA to improve punching has been examined in MMA (Table 2) [75]. Consuming 5 mg/kg of caffeine 60 min before the test did not alter punch frequency, punch force or perceptual training intensity compared with the placebo group.

A long protocol with 10 g/day of sodium bicarbonate (21 days) before the Wingate test in non-specified combat sports increased total work (+10.9%) and peak (+10.7%) and mean (+11.4%) power only in the upper limbs [76]. Blood lactate levels were higher (+13.7%) than the placebo group while other biochemical parameters such as insulin-like growth factor 1 (IGF-1) and cortisol increased and decreased, respectively.

In a study in which the authors did not specify the combat disciplines, participants were given a beetroot-based gel containing 12.2 mmol of NO_3_^−^ 120 min before a forearm muscle isometric strength test and handgrip isotonic exercise [77]. Compared with the placebo group, there was a significant increase in maximal voluntary forearm contraction but not changes in muscle oxygen saturation, time until fatigue and blood volume in the forearm.

### 3.7. Caffeine Meta-Analysis

When the effect of caffeine consumption was analysed, the placebo group showed significantly lower levels of lactate (Figure 2; *p* < 0.0001).

## 4. Discussion

### 4.1. Effects of Caffeine in Combat Sports

Caffeine exerts its role as an agonist of adenosine A1 and A2a receptors [78], modulating central nervous system activity by inhibiting parasympathetic activity. At the metabolic level, caffeine leads to elevated blood norepinephrine levels, enhancing glycolytic activity to increase muscle energy supply during high-intensity exercise [79,80,81]. Caffeine is an effective ergogenic aid for aerobic and anaerobic exercise, providing improvements in performance and the perceptions of exertion and muscle pain with doses ranging from 2.35 to 5 mg/kg [82,83]. Coffee and derivates are habitual drinks in many cultures around the world, so it is common for athletes to consume caffeine from these sources. It does not seem that habitual caffeine consumption affects the ergogenic effects of caffeine [84].

Combat sports involve a multitude of muscle groups and high-intensity intermittent actions, due to the large number of attacks with great force and speed that utilise the energy provided from the anaerobic metabolic pathway. Although studies indicate that the aerobic energy pathway is the main one and contributes to the recovery process during breaks between rounds, the decisive actions are maintained by anaerobic processes [85,86,87,88,89,90]. In combat sports, most studies have analysed caffeine doses from 5 to 10 mg/kg and have reported improvements in various aspects of performance. In both specific and general skills, the improvements are based mainly on abilities related to glycolytic metabolism: the effort–pause ratio, the time of punching sequences, the number of attacks, reaction and skipping time, handgrip strength, power, static lifts and TTE, among others [24,25,26,27,28,30,49,52,58]. More studies that evaluate the combination of caffeine and sodium bicarbonate are necessary to verify a clear synergistic response, because the available results are heterogenous [27,54] as well as compared to the conclusion reached in other studies [91]. In this regard, athletes are able to reach higher intensity levels due to a greater production and tolerance of blood lactate levels (2.1–20.9 vs. 3.2–18.9 mmol/L in the caffeine and placebo groups, respectively) through the use of the glycolytic pathways for energy production (Figure 2). These findings are similar to results observed in disciplines where glycolytic metabolism is involved [92,93,94].

The results that caffeine consumption increased blood lactate levels compared with the placebo group are consistent with the results of a recent review [95]. Although this review explained that caffeine increases the blood lactate concentration, it is necessary to consider that caffeine ingestion is used to enhance an athlete’s physical performance, and it could allow them to produce more intense efforts for a longer time. This sustained effort could lead to higher blood lactate levels. On the other hand, the perception of effort was improved when using a range of 4–10 (5 × 2) mg/kg of caffeine in only five studies [24,25,42,48,52], with a fair score on the PEDro scale for one of them [25]. In 12 studies that provided the participants with 3–9 mg/kg caffeine, there was no difference compared with the placebo group, or even an increased feeling of fatigue coinciding with the lowest doses [27,28,29,30,49,50,53,54,55,57,58,75]. These findings are consistent with results in other disciplines [96]. In 1- and 3-day protocols with caffeine and sodium bicarbonate, caffeine did not improve perceived exertion [27,54]. Additional studies are necessary with higher doses of caffeine to determine whether there is a strong effect in the effort perception during high-intensity tasks.

### 4.2. Effects of Buffering Supplements in Combat Sports

Bicarbonate coming from carbon dioxide (CO_2_) acts as the main mechanism to buffer plasma acidification. Normally, a drop in muscle and plasma pH occurs during high-intensity exercise, because acid (H^+^) and CO_2_ tend to accumulate [97]. The efficacy of acute sodium bicarbonate supplementation is influenced by the duration of exercise. Specifically, sports of prolonged duration (>4 min) have shown mixed results with the use of sodium bicarbonate supplementation, improving performance in running and cycling, but not in rowing, rugby, water polo or basketball [97]. Because sodium bicarbonate could cause gastrointestinal discomfort [98], other buffer supplements such as sodium citrate have been tested. On the other hand, beta-alanine acts as intracellular buffer, increasing carnosine content and, subsequently, improving high-intensity exercise capacity in cycling [99]. A meta-analysis revealed improved high-intensity endurance performance from 30 s to 10 min in duration [100].

An acute dose of 0.3 g/kg of sodium bicarbonate for 1 or 3 days improved the number of throws, power and TTE in combat sports [27,32,33,34,63]. This finding is similar to studies with racquet sports, where specific skills and TTE tended to improve [94]. On the other hand, the same protocol could not improve handgrip strength and total forearm contractions [59]. Lower doses had little to no effect on the number of throws (until 0.1 g/kg) [54] and the power (in a protocol of progressive dose increase from 0.025 g/kg to 0.1 g/kg over 10 days) [60,61]. Despite improvements in punch efficacy, attack time and total work used for sports [27,32,33,34], additional studies are needed with more participants and a more extensive dose range to verify this fact. Regarding physiological aspects, there was an increase in blood lactate, without affecting heart rate or fatigue perception, using high doses (0.3 and 0.5 g/kg 60–90 min but not 120 min before the exercise or 10 g/day for 1 week) in different protocols [32,33,62,63,76] but mixed results were obtained with lower doses [54,60,61]. An exception was the decrease in blood lactate after a taekwondo-specific test with 0.5 g/kg/day of sodium bicarbonate for 5 days with or without creatine [31]. The significant increase in blood lactate levels compared with the placebo group could be due to carboxylate co-transporter, which extracts lactate and H^+^ from working muscle cells to the circulation after an increase in extracellular pH [101] and an increase in glycolytic activity. It seems that only doses ≥ 0.3 g/kg sodium bicarbonate produced an increase in pH and glycolytic energy contribution, but more studies using a wide range of doses are necessary to determine the optimal dose and to standardise exercise protocols.

Protocols using sodium citrate showed similar results regarding blood lactate (but only with one study using 0.9 g/kg for one day [64]) and pH increase (with a 1-day protocol providing 0.6–0.9 g/kg [64,65]). These results are similar to those observed with lower doses (0.5 g/kg) in intermittent sports such as tennis [92] and in swimmers using 0.3 g/kg in a 400 m time-trial test [93]. While in tennis, but not in swimming, some specific skills were improved, in combat sports they were not tested [64,65,92,93]. Sodium citrate has also demonstrated improvements in water intake retention and plasma volume (without affecting dehydration parameters measured in urine) [65], perhaps due to its influence on hormone diuresis control [102]. Similar to sodium bicarbonate, there were no changes compared with the placebo group in terms of heart rate or fatigue perception.

Intracellular buffering using 4.9–6.4 g/day of beta-alanine for 4–10 weeks in combat sports improved power in general physical tests for both striking and grappling disciplines [36,62]. On the other hand, strength, total work or running time showed less evidence of benefits [35,36,62,67]. Specific combat skills, such as punch efficacy and the number of throws, tended to improve with beta-alanine supplementation (6.0–6.4 g/day for 4 weeks) [37,66]. However, more studies are necessary to obtain stronger evidence, because other athletes such as climbers—who perform a large amount of isometric work with forearm muscles, similar to grappling fighters—showed improvements in repeated high-intensity intermittent upper body performance with lower doss (4 g/d during 4 weeks) [103]. Regarding physiological improvements with beta-alanine supplementation, mixed results have been obtained regarding blood lactate levels [36,37,62,66,67]. There were no changes in blood pH [66] or fatigue perception when beta-alanine was used alone—but fatigue perception was enhanced when used together with sodium bicarbonate [62].

Buffering supplements appear to have greater benefits in grappling disciplines, but additional investigation is necessary with the aim of achieving strong evidence regarding the optimal dose range that positively affects specific combat skills, as well as its possible synergistic effects with the combination of several of them and other NEAs (i.e., caffeine).

### 4.3. Effects of Creatine Monohydrate in Combat Sports

Creatine monohydrate supplementation has been used as a strategy to increase strength and muscle mass during training, but it has also been reported to improve power and anaerobic capacity [104,105,106]. Thus, the use of creatine in combat sports such as judo is of great interest because about 10% of elite Japanese and Korean judoists take it [107].

The use of high doses of creatine (0.3 g/kg/day or 20 g/day) for 5–15 days improved performance (agility and power) [31,69] and combat skills in grappling disciplines (handgrip strength) [69], but the study obtained a fair score for the PEDro scale. The results are in the line with other intermittent disciplines such as racquet sports, where only 0.3 g/kg for 5 days improved sprint time in squash players [94]. Additional studies with specific-combat tests, better-quality experimental design (similar baseline sample inclusion criteria and blinding subjects, therapists and assessors regarding the treatments) and greater homogeneity in the protocols are necessary to confirm this fact with a wide range of doses and in striking disciplines. At present, there is no evidence for its recommendation.

### 4.4. Effects of Nitric Oxide Precursors in Combat Sports

NO has a relevant role as an intracellular second messenger and its production is also related to an increase in blood flow, which improves nutrient and hormone delivery. Furthermore, NO has a positive impact on resistance and endurance training adaptions [108,109]. Recent systematic reviews and meta-analyses about NO synthase–independent pathway supplementation have shown that sodium nitrate and potassium nitrate are less effective than beetroot juice consumption in endurance exercise. The use of 6–12 mmol of NO_3_^−^ contained in beetroot juice supplements produced significant improvements in time to exhaustion in a 5–30 min cycling race, but slightly non-significant improvements in time trial or graded-exercise performance [80]. Regarding combat sports, neither 2 g/kg of beetroot juice [41] nor 400 and 800 mg of NO_3_^−^ contained in the juice [39] produced performance advantages; indeed, there were negative effects on power compared with the placebo group [41]. Misinformation regarding the NO_3_^−^ concentration of the product [41] did not permit determining whether the ergogenic dose established in other sports trials was achieved. On the other hand, beetroot-based gel supplementation for 8 days or 120 min before exercise increased maximal voluntary forearm contraction and muscle oxygen saturation and decreased blood lactate levels after exercise (improvements in lactate clearance) compared with the placebo group in an isometric test exercise [71,77]. In this case, the NO_3_^−^ concentration was 12.2 mmol, near the upper end of the range established for other studies with improvements in performance [110,111]. Beetroot extract (without information on the NO_3_^−^ content) taken before exercise improved VO_2_ peak and the anaerobic threshold without affecting blood lactate levels [40]. More studies are necessary using beetroot-based products and well-established nitrate-containing beetroot juices in specific combat tests before providing adequate advice for combat sports.

In the case of NO synthase–dependent pathway, supplementing l-arginine as its natural precursor is not classified as a group A aid by the AIS, but recent publications have shown interesting data that could support its use [112]. Anaerobic performance, the main pathway to obtain energy during high-intensity actions in combat sports, is enhanced by acute (0.15 g/kg of l-arginine 60–90 min before exercise) and chronic (10–12 g/day for 8 weeks) use [112]. Only an acute protocol with 150 mg/kg of l-arginine 60 min before exercise enhanced TTE in wrestlers [72], according to meta-analyses results [112], but the study obtained a fair score on the PEDro scale. A chronic protocol with a lower dose (6 g/day of l-arginine) than the range established as ergogenic and taken for less time (3 days) had no effects in judoists [73]. In this sense, it is too early to establish a robust recommendation regarding l-arginine supplementation.

### 4.5. Effects of Glycerol Supplementation in Combat Sports

Glycerol is a metabolite that acts as a plasma expander and could help athletes maintain euhydration and improve thermoregulatory and cardiovascular changes [9]. Until 2018, WADA had considered glycerol a banned substance because it was believed that it could alter an athlete’s biological passport [113]. In any case, the results of its supplementation are mixed, both in endurance and anaerobic disciplines [8]. In combat sports, 1.0 g/kg glycerol before the Wingate test in wrestlers could not improve anaerobic power and did not affect body mass, urine specific gravity or saliva osmolality [74]. More research is needed to determine glycerol’s supposed potential efficacy in combat sports in specific tests and in more disciplines.

## 5. Conclusions

Caffeine is the NEA for which there is clear evidence of benefits for combat sports practitioners. Acute doses (5–10 mg/kg) 30–60 min before combat may improve specific skills that rely on glycolytic metabolism to obtain energy, but it may not contribute to improve perceived exertion. Even though some evidence concludes that other NEAs show promise in improving performance, such as buffering supplements, more studies are necessary, specifically for grappling disciplines, to verify its validity during sustained isometric efforts. Creatine, NO precursors or glycerol could play an interesting role in improving performance, but more studies are needed to strengthen the evidence.

## Figures and Tables

**Figure 1 nutrients-14-02588-f001:**
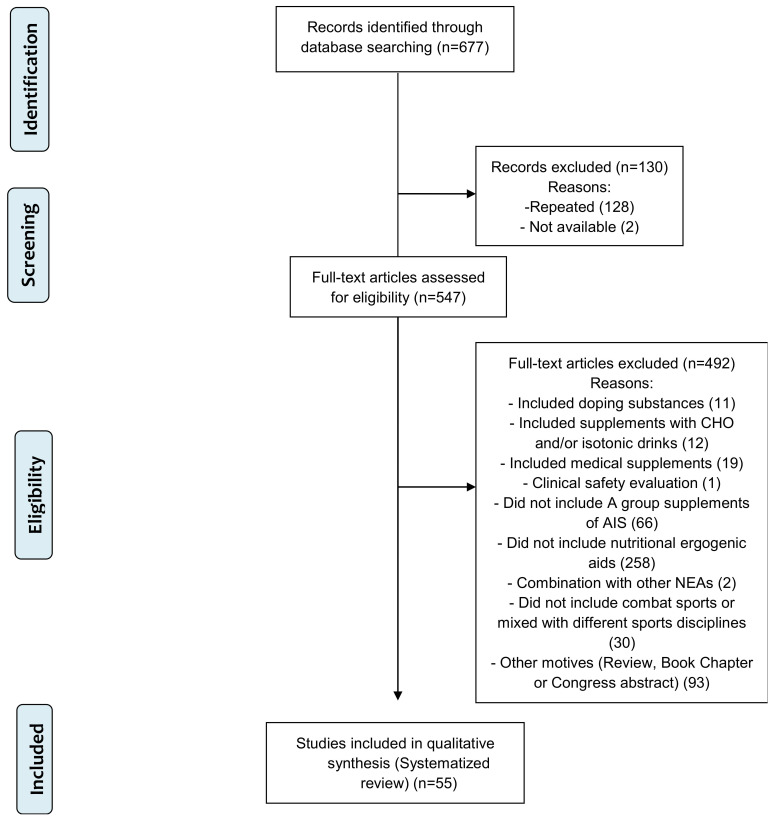
PRISMA flow chart [16] of the study selection process.

**Figure 2 nutrients-14-02588-f002:**
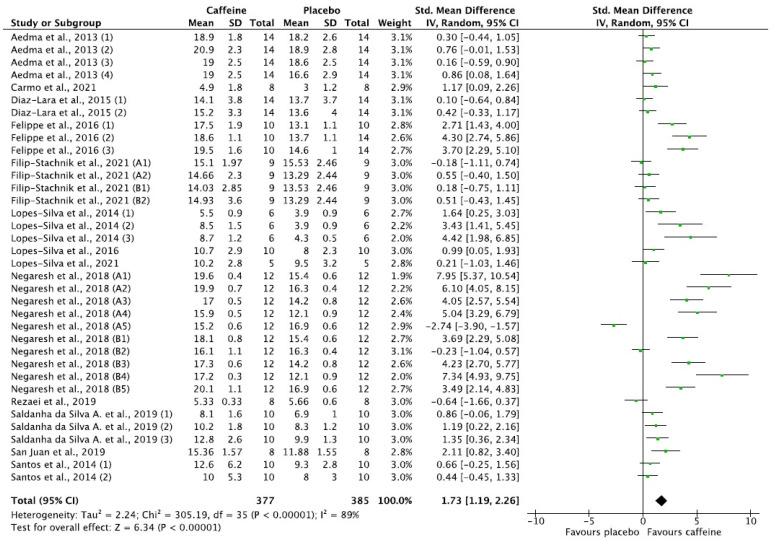
Forest plot results of a random effects meta-analysis for the placebo group compared with the experimental (caffeine) group [26,27,29,30,45,46,47,48,50,54,55,56,57].

**Table 1 nutrients-14-02588-t001:** Combined Mesh terms used in the search of studies in database. ^1^: NEAs filed in group A of AIS.

NEA ^1^		Sport
Dietary supplements	AND	Martial Arts
Caffeine		Boxing
Creatine		Wrestling
Beta-alanine		
Sodium Bicarbonate		
Nitrates		
Glycerol		

**Table 2 nutrients-14-02588-t002:** Included studies of nutritional ergogenic aids in combat sports. 1RM: one-repetition maximum; BA: beta-alanine; BDNF: brain-derived neurotrophic factor; CAF: caffeine; CMJ: countermovement jump; EMG: electromyography; FFA: free fatty acids; FSKT: frequency speed of kick test multiple; HIIR: high intensity interval run; HS: handgrip maximal strength; IGF-1: insulin-like growth factor 1; JGST: judogi grip strength test; KSAT: karate-specific aerobic test; MMA: mixed martial arts; PSTT: progressive specific taekwondo test; PWPT: Pittsburgh wrestling performance test; SB: sodium bicarbonate; SFJT: special fitness judo test; TAIKT: taekwondo anaerobic intermittent kick test; TSAT: taekwondo-specific agility test; TTE: time to exhaustion; UBISP: upper-body intermittent sprint performance test.

Study	NEA	Dosage/Time	Duration	Participants (Gender)	Age(yrs)	Combat Sport	Level	Exercise Protocol	Main Outcomes
Striking Combat Sports
[23] Ouergui et al. (2022)	Caffeine	3 mg/kg, 60 min before test	1 day	20 (10 male/10 female)	17.5 ± 0.7	Taekwondo	?	TSAT + FSKT	↓ Time of agility test ↑ Total number of kicks  Perceptual training intensity  Mood, feeling and vitality
[24] Jodra et al. (2020)	Caffeine	6 mg/kg, 60 min before test	1 day	8 (male)	22.0 ± 1.8	Boxing	International-level	Wingate test	↑ Power  Perceptual training intensity↑ Tension, vigor and vitality↓ Fatigue
[25] Pak et al. (2020)	Caffeine	6 mg/kg (mouth rinsing), 0 min before test	1 day	27 (18 male/9 female)	17.0 ± 3.0	Taekwondo	State-level	TAIKT test before, during and after Ramadan period (fed and fasting comparison)	↑ % Successful kicks (during the first 3 weeks of Ramadan)↓ Perceptual training intensity (during all weeks of Ramadan)
[26] San Juan et al. (2019)	Caffeine	6 mg/kg, 60 min before test	1 day	8 (male)	22.0 ± 1.8	Boxing	International-level	Wingate test + EMG + CMJ + HS	↑ Power↑ Jump height  Jump power  Handgrip maximal strength  Blood lactate
[27] Rezaei et al. (2019)	Caffeine	6 mg/kg, 50 min before test	1 day	8 (?)	20.5 ± 2.4	Karate	State-level	KSAT	↑ TTE  Vertical jumps (high)  Blood lactate  HR  Perceptual training intensity
[28] Coswig et al. (2018)	Caffeine	6 mg/kg, 30 min before test	1 day	10 (male)	25.9 ± 5.2	Boxing	Amateur	Combats of 3 × 2 min	↑ Effort–Pause ratio↑ Time of punching sequences (round 1 and 2)  Number of punching sequences  HR  Perceptual training intensity
[29] Lopes-Silva et al. (2015)	Caffeine	5 mg/kg, 60 min before test	1 day	10 (male)	21.0 ± 4.0	Taekwondo	International-level	Combat of 3 × 2 min	 Attack Time  Total number of attacks  Stepping time↑ Blood lactate  HR  HR variability  HR recovery  Time-varying vagal-related index  VO_2_  Aerobic energy contribution  ATP-PCr energy contribution↑ Glycolytic energy contribution  Energy expenditure  Perceptual training intensity
[30] Santos et al. (2014)	Caffeine	5 mg/kg, 50 min before test	1 day	10 (male)	24.9 ± 7.3	Taekwondo	Amateur	Combats of 2 × 3 × 2 min + Reaction-time Test before, between and after the 2 combats	↓ Reaction time before combat 1 ↑ Number of attacks in combat 2↓ Number of referee breaks in combat 1 ↓ Skipping times in combat 2 ↑ Blood lactate in combat 1  HR  Perceptual training intensity
[31] Sarshin et al. (2021)	Sodium Bicarbonate	-SB: 4 × 0.125 g/kg/day	5 days	40 (male)	21.4–23.1 ± 1.1–2.4	Taekwondo	National level	TAIKT	↑ Peak and Mean Power↓ Blood lactate after test
[32] Gough et al. (2019)	Sodium Bicarbonate	0.3 g/kg, 65 min before test	1 day	7 (male)	27.1 ± 5.1	Boxing	International-level	HIIR + Punch test + HIIR	↑ TTE↑ Blood lactate after 2nd HIIR↑ pH after 1st HIIR↑ HR in Punch test  Perceptual training intensity
[27] Rezaei et al. (2019)	Sodium Bicarbonate	-0.3 g/kg/day before test day-0.1 g/kg, 120, 90 and 60 min before test	3 days	8 (?)	20.5 ± 2.4	Karate	State-level	KSAT	↑ TTE  Vertical jumps (high)  Blood lactate  HR  Perceptual training intensity
[33] Lopes-Silva et al. (2018)	Sodium Bicarbonate	0.3 g/kg, 90 min before test	1 day	9 (male)	19.4 ± 2.2	Taekwondo	National-level	Combat of 3 × 2 min	↑ Attack Time  Total number of attacks  Stepping time↑ Blood lactate  HR  VO2  Aerobic energy contribution  ATP-PCr energy contribution↑ Glycolytic energy contribution in round 1  Energy expenditure  Perceptual training intensity
[34] Siegler et al. (2010)	Sodium Bicarbonate	0.3 g/kg, 60 min before test	1 day	10 (?)	22.0 ± 3.0	Boxing	Amateur	Combat of 4 × 3 min	↑ Punch efficacy  HR in Punch test  Perceptual training intensity
[27] Rezaei et al. (2019)	Sodium Bicarbonate + Caffeine	-SB: 0.3 g/kg/day before test day-SB: 0.1 g/kg, 120, 90 and 60 min before test-CAF: 6 mg/kg, 60 min before test	3 days	8 (?)	20.5 ± 2.4	Karate	State-level	KSAT	↑ TTE  Vertical jumps (high)  Blood lactate  HR  Perceptual training intensity
[31] Sarshin et al. (2021)	Sodium Bicarbonate + Creatine	-SB: 4 × 0.125 g/kg/day-CRE: 4 × 5 g/day	5 days	40 (male)	21.4–23.1 ± 1.1–2.4	Taekwondo	National level	TAIKT	↑ Peak Power↑ Mean Power and > SB or CRE alone↓ Blood lactate after test
[35] Alabsi et al. (2022)	Beta-alanine	20.7–24.4 g/day (0.3 g/kg)	4 weeks	18 (male)	22.0–24.4 ± 4.7–5.8	Boxing	-	Strength training + Wingate test	 Peak and Mean Power  Fatigue Index  Blood lactate
[36] Kim et al. (2018)	Beta-alanine	4.9–5.4 g/day (3 × 1650–1800 mg/day)	10 weeks	19 (male)	22.2–23.0 ± 2.2–1.8	Boxing	Amateur	Physical fitness	 Maximal strength  Isokinetic strength↑ Peak power lower limbs  Mean power  Power endurance↓ Power drop upper limbs  Blood lactate
[37] Donovan et al. (2012)	Beta-alanine	6 g/day (4 × 1500 mg/day)	4 weeks	16 (?)	25.0 ± 4.0	Boxing	Amateurs	Simulated boxing protocol with a punch bag of 3 × 3 min	↑ Number of punches↑ Mean and accumulative punch force↑ Blood lactate  HR
[31] Sarshin et al. (2021)	Creatine	4 × 5 g/day	5 days	40 (male)	21.4–23.1 ± 1.1–2.4	Taekwondo	National level	TAIKT	↑ Peak and Mean Power  Blood lactate after test
[38] Manjarrez-Montes de Oca et al. (2013)	Creatine	50 mg/kg/day	6 weeks	10 (male)	20.0 ± 2.0	Taekwondo	Amateur	Wingate test	 Peak and Mean Power  Fatigue Index  Blood lactate ↑ Triglycerides
[39] Miraftabi et al. (2021)	Beetroot juice	-400 mg NO_3_^−^, 150 min before test-800 mg NO_3_^−^, 150 min before test	1 day	8 (male)	20.0 ± 4.0	Taekwondo	National level	CMJ + FSKT + Rest + CMJ + PSTT	 Total number of kicks  TTE  Jump height, flight time, velocity force and power  Blood lactate  Anaerobic performance (kick decrement index)  HR  Perceptual training intensity
[40] Antonietto et al. (2021)	Beetroot extract	1 g/? before test	1 day	12 (male)	26.8 ± 8.8	Taekwondo	-	PSTT	 Blood lactate ↑ VO2 peak↑ Anaerobic threshold
[41] Tatlici et al. (2019)	Beetroot juice	2 g/kg, 150 min before test	1 day	8 (male)	23.0 ± 2.3	Boxing	International and National	Upper body Wingate test	↓ Peak and Mean Power  Blood lactate  HR
Grappling combat sports
[42] Merino Fernández et al. (2022)	Caffeine	3 mg/kg, 60 min before test	1 day	22 (11 male/11 female)	22.0 ± 4.0	Jiu-jitsu	?	SFJT + Combats	↑ Total throws↑ SFJT index  Number of attack and defensive actions ↑ HR↑ Strength and Endurance perception↓ Fatigue perception
[43] Krawczyk et al. (2022)	Caffeine	3 and 6 mg/kg, 60 min before test	1 day	16 (6 male/4 female)	Male: 26.4 ± 5.3Female: 20.8 ± 1.5	Judo	National-level	3 x3 Bench-press + 3 × 3 Bench-pull + CMJ + Handgrip strength test + JGST	↑ Mean velocity Bench-press (only with 6 mg/kg) ↑ Peak velocity Bench-press ↑ Mean velocity Bench-pull  Peak velocity Bench-pull  Jump height ↑ Number of repetitions of Grip strength  Grip endurance strength  Handgrip maximal strength
[44] Merino Fernández et al. (2021)	Caffeine	3 mg/kg, 60 min before test	1 day	16 (8 male/8 female)	Male: 21.5 ± 4.75Female: 20.63 ± 3.20	Jiu-jitsu	?	Bilateral and Unilateral CMJ	↑ Power (bilateral)↑ Jump height (bilateral and right leg)↑ Flight time (bilateral and right leg)
[45] Lopes-Silva et al. (2021)	Caffeine	5 mg/kg, 60 min before test	1 day	10 (?)	25.2 ± 5.3	JudoJiu-jitsu	National level	4xJudogi’s dynamic strength endurance test + 4xHandgrip force after each bout	↑ Total number of repetitions↑ Maximum Isometric Handgrip strength  Blood Lactate  HR  Perceptual training intensity
[46] Filip-Stachnik et al. (2021)	Caffeinated chewing gum	2.7 and 5.4 mg/kg, 15 min before each SFJT test	1 day	9 (male)	23.7 ± 4.4	Judo	International and National level	SFJT + Combats of 4 min + SFJT	 Total throws  SFJT index (HR/Total throws)  Blood Lactate  HR  Perceptual training intensity
[47] Carmo et al. (2021)	Caffeine	5 mg/kg, 60 min before test	1 day	8 (male)	21.4 ± 2.0	Judo	National level	SFJT + CMJ + Upper limb power test + General exercises (40 min) + Technical training (40 min) + Combats of 8 × 4 min + SFJT + CMJ + Upper limb power test	↑ Total throws in post-training↓ Fatigue index in post-training  Power upper limbs↑ Plasma FFA at 120 min↓ Serum Uric acid at 120 min  erum Creatinine and Glucose at 120 min↑ Blood Lactate at 120 min  Urine production  HR  Blood pressure  Perceptual training intensity
[48] Negaresh et al. (2019)	Caffeine	-4 and 10 mg/kg, 45 min before first combat-5 × 2 mg/kg, 45 and 30 min before each combat-6.2 mg/kg, 30 min before first combat	1 day	12 (male)	24.0 ± 3.0	Wrestling	Professional	PWPT-Hip/back strength-Vertical jump + 5 Combats of 2 × 3 min (PWPT-Hip/back strength-Vertical jump before each combat) + Hip/back strength-Vertical jump	 Hip/back strength  Jump Height↓ Time to complete PWPT (only with 5 × 2 and 6.2 mg/kg of caffeine) ↑ Blood lactate after 3rd combat (only with 5 × 2 and 6.2 mg/kg of caffeine); before 4th combat (only with 5 × 2 mg/kg) and after 4th and 5th combat (only with 6.2 mg/kg)↓ Blood lactate before 4th combat (only with 6.2 mg/kg)  Urine osmolality  Urine specific gravity↑ Dehydration index (only with 10 mg/kg)↑ HR before 2nd combat (only with 10 mg/kg); after the 3rd combat (only with 4 mg/kg and 5 × 2 mg/kg)↓ HR after 5th combat (only with 6.2 mg/kg)↓ Perceptual training intensity (only with 5 × 2 and 6.2 mg/kg of caffeine)
[49] Durkalec-Michalski et al. (2019)	Caffeine	6–9 mg/kg, 60 min before test	1 day	22 (male)	21.7 ± 3.7	Judo	State-level	SFJT and judo sparring combats (Randori)	↑ Total throws of opponent (higher at 9 than 6 mg/kg)  SFJT index (HR/Total throws)↑ Total attacks in combat↑ HR  Perceptual training intensity
[50] Saldanha da Silva et al. (2019)	Caffeine	5 mg/kg, 60 min before test	1 day	12 (male)	23.1 ± 4.2	Judo	State-level	Combats of 3 × 5 min	 Total number of attacks  Efficiency or effectiveness scores  Perceived recovery  Perceptual training intensity
[51] Athayde et al. (2018)	Caffeine	5 mg/kg, 60 min before test	1 day	14 (male)	22.5 ± 7.1	Judo	State-level	Combats of 3 × 5 min + CMJ-HS-JGST between combats	 Total number of attacks  Jump height and power  Grip endurance strength  Handgrip maximal strength↑ Blood lactate
[52] Astley et al. (2017)	Caffeine	4 mg/kg, 60 min before test	1 day	18 (male)	16.1 ± 1.4	Judo	State-level	SFJT	↓ SFJT index (HR/Total throws)↑ Number of throws  HR↓ Perceptual training intensity
[53] Diaz-Lara et al. (2016 a)	Caffeine	3 mg/kg, 60 min before test	1 day	14 (male)	29.2 ± 3.3	Brazilian Jiu-jitsu	National-level	Handgrip force + CMJ + Maximal static lift + 1RM + Bench-press repetitions to failure	↑ Handgrip maximal strength↑ Maximum static lift↑ Jump height↑ Velocity at peak power in jumps  Force applied at peak power in jumps↑ Weight, power and velocity in 1RM↑ Number of bench-press repetitions↑ Perceptual training intensity
[54] Felippe et al. (2016)	Caffeine	6 mg/kg, 60 min before test	1 day	10 (male)	23.0 ± 5.0	Judo	National-level	SFJT	 Number of throws↑ Blood lactate  Perceptual training intensity
[55] Diaz-Lara et al. (2016 b)	Caffeine	3 mg/kg, 60 min before test	1 day	14 (male)	29.2 ± 3.3	Brazilian Jiu-jitsu	National-level	Combats of 2 × 8 min + (1RM in Bench-press + HS+ CMJ + Maximal static lift) before, between and after the 2 combats	↑ Number of high-intensity offensive actions  Number of defensive actions↑ Handgrip maximal strength before combats↑ Maximum static lift beforecombats and post-combat 1↑ Jump height before combats↑ Power in 1RM before combats and post-combat 1↑ Velocity in 1RM  Blood lactate pre- and post-combat 1↑ Blood lactate pre- and post-combat 2↑ Perceptual training intensity
[56] Lopes-Silva et al. (2014)	Caffeine	6 mg/kg, 60 min before test	1 day	6 (male)	25.3 ± 5.7	Judo	National-level	Reduction of 5% body weight for 5 days + SFJT	 Number of throws  SFJT index (HR/Total throws)↑ Blood lactate  HR↓ Perceptual training intensity
[57] Aedma et al. (2013)	Caffeine	5 mg/kg, 30 min before test	1 day	14 (?)	25.3 ± 4.9	Brazilian jiu-jitsu and Wrestling	Amateur	UBISP	 Power↑ Blood lactate ↑HR↑HR recovery  Perceptual training intensity
[58] Souissi et al. (2012)	Caffeine	5 mg/kg, 60 min before test	1 day	12 (?)	21.1 ± 1.2	Judo	?	Reaction time test + Wingate test	↓ Reaction time ↑ Peak and Mean Power  Fatigue Index↑ Vigor and Anxiety
[59] Ragone et al. (2020)	Sodium Bicarbonate	3 × 0.1 g/kg, 80, 70 and 60 min before test	1 day	10 (male)	22.2-± 3.9	Jiu-jitsu	National-level	Handgrip strength test + Forearm muscle intermittent isometric contraction test	 Maximum and Mean Handgrip strength  Number of contractions  Total time of contractions  Blood lactate↑ pH
[60] Durkalec-Michalski et al. (2020)	Sodium Bicarbonate	-−0.025 g/kg/days 1–2-−0.05 g/kg/days 3–5-−0.075 g/kg/days 6–7-−0.1 g/kg/day 8–10	10 days	51 (33 male/18 female)	Male: 19.5–19.7 ± 3.8–4.4Female: 18.1–18.7 ± 2.4–2.6	Wrestling	National-level	Wingate test + Dummy throw test + Wingate test	 Peak and Mean Power↑ Difference in power indices between 2nd and 1st Wintgate test (in 12, 16, 17 and 21 s)  Number of throws  Blood lactate  Blood glucose  Blood pyruvate
[61] Durkalec-Michalski et al. (2018)	Sodium Bicarbonate	-−0.025 g/kg/days 1–2-−0.05 g/kg/days 3–5-−0.075 g/kg/days 6–7-−0.1 g/kg/day 8–10	10 days	49 (31 male/18 female)	18.0–19.0 ± 4.0	Wrestling	National-level	Wingate test + Dummy throw test + Wingate test	↓ Time to peak power  Peak, Mean and Minimum Power  Number of throws  Blood lactate  Blood glucose
[54] Felippe et al. (2016)	Sodium Bicarbonate	0.1 g/kg, 120, 90 and 60 min before test	1 day	10 (male)	23.0 ± 5.0	Judo	National-level	SFJT	 Number of throws↑ Blood lactate  Perceptual training intensity
[62] Tobias et al. (2013)	Sodium Bicarbonate	0.5 g/kg (4 × 12 mg/kg)	7 days	37 (male)	23.0 ± 4.0	JudoJiu-jitsu	International, National and State-level	4 bouts of Wingate test	↑ Total work↑ Peak and Mean Power in 4th bout↑ Blood lactate  Perceptual training intensity
[63] Artioli et al. (2007)	Sodium Bicarbonate	0.3 g/kg, 120 min before test	1 day	23 (?)	19.3–21.5 ± 2.4–3.0	Judo	International and National-level	3 bouts of SFJT (*n* = 9)4 bouts of Wingate test (*n* = 14)	↑Number of throws↑ Peak and Mean Power in 4th bout of Wingate test↑ Blood lactate in 3rd bout of SFJT  Blood lactate in Wingate test  Perceptual training intensity
[64] Aedma et al. (2015 a)	Sodium Citrate	0.9 g/kg, 16 h, 8 h (aprox.) and 30 min before test	1 day	11 (?)	25.9 ± 6.2	Brazilian jiu-jitsu and Wrestling	?	4 UBISP tests (4 × 6 min)	 Peak and Mean Power↑ pH↑ Blood lactate after 1st test  Urine osmolality  Urine specific gravity  Urine volume↑ Water intake and retention↓ Decreasing in plasma volume  HR  Perceptual training intensity
[65] Timpmann et al. (2012)	Sodium Citrate	0.6 g/kg, 16 h, 8 h (aprox.) and 120 min before test + rapid body mass loss	1 day	16 (?)	22.5 ± 3.9	Wrestling	?	UBISP	 Peak and Mean Power↑ pH  Blood lactate  Urine specific gravity↑ Increasing in plasma volume  Perceptual training intensity
[54] Felippe et al. (2016)	Sodium Bicarbonate + Caffeine	-SB: 0.1 g/kg, 120, 90 and 60 min before test-CAF: 6 mg/kg, 60 min before test	1 day	10 (male)	23.0 ± 5.0	Judo	National-level	SFJT	↑ Number of throws↑ Blood lactate  Perceptual training intensity
[66] de Andrade Kratz et al. (2017)	Beta-alanine	6.4 g/day (4 × 1600 mg mg/day)	4 weeks	23 (male)	17.2–19.3 ± 2.0–3.0	Judo	International and National	3 bouts of SFJT	↑ Number of throws  pH  Blood lactate
[62] Tobias et al. (2013)	Beta-alanine	6.4 g/day (4 × 1600 mg mg/day)	4 weeks	37 (male)	26.0 ± 4.0	JudoJiu-jitsu	International, National and State-level	4 bouts of Wingate test	↑ Total work↑ Mean Power in 2nd and 3rd bout  Peak Power↑ Blood lactate  Perceptual training intensity
[67] Kern et al. (2011)	Beta-alanine	4.4 g/day (2 × 2200 mg mg/day)	8 weeks	22 (male)	19.9 ± 1.9	Wrestling	Amateurs	Running test (274 m) + Time of hanging 90º elbows flexed	 Time running  Flexed arm hang time  Blood lactate
[62] Tobias et al. (2013)	Beta-alanine + Sodium Bicarbonate	-BA: 6.4 g/day (4 × 1600 mg mg/day)-SB: 0.5 g/kg (4 × 12 mg/kg)	-BA: 4 weeks-SB: 7 days	37 (male)	26.0 ± 5.0	JudoJiu-jitsu	International, National and State-level	4 bouts of Wingate test	↑ Total work and > Ba or SB alone↑ Mean Power in all bouts↑ Peak Power in 1st, 2nd and 3rd bouts↑ Blood lactate ↓ Perceptual training intensity
[68] Aedma et al. (2015 b)	Creatine	0.3 g/kg/day (4 × 75 mg/kg/day)	5 days	20 (male)	25.6 ± 3.8	Brazilian jiu-jitsu and Wrestling	Amateur	UBISP	 Peak and Mean Power  Urine specific gravity  Blood lactate  HR  HR recovery  Perceptual training intensity
[69] Abbasalipour et al. (2012)	Creatine	0.3 g/kg/day	15 days	14 (?)	18.0–25.0	Wrestling	Amateur	Agility test + Handgrip strength test	↑ Handgrip strength↑ Agility
[70] Tatlici (2021)	Beetroot juice	140 mL (600 mg NO_3_^−^), 150 min before test	1 day	8 (male)	21.9 ± 2.3	Wrestling	-	Knee isokinetic strength test + Shoulder internal and external rotation isokinetic strength test	 Peak isokinetic strength lower limbs↑ Peak isokinetic strength upper limbs↑ Mean isokinetic strength lower and upper limbs
[71] de Oliveira et al. (2018)	Beetroot-based gel	12.2 ± 0.2 mmol Nitrate	8 days	12 (male)	29.0 ± 9.0	Brazilian jiu-jitsu	Amateur	Forearm muscle isometric strength test + Handgrip isotonic exercise	↑ Maximal voluntary forearm contraction↑ Muscle O_2_ saturation during exercise recovery  Blood volume in forearm  Time until fatigue ↓ Blood lactate post-exercise
[72] Yavuz et al. (2014)	Arginine	150 mg/kg, 60 min before test	1 day	9 (male)	24.7 ± 3.8	Wrestling	International and National	Incremental cycloergometer test to exhaustion	↑ TTE  Blood lactate  HR  VO2
[73] Liu et al. (2009)	Arginine	6 g/day	3 days	10 (male)	20.2 ± 0.6	Judo	International and National	Intermittent anaerobic exercise test	 Peak and Mean Power  Blood lactate, ammonia, citrulline, nitrate and nitrite during and post-exercise↑ Blood arginine during and post-exercise
[74] McKenna et al. (2017)	Glycerol	1 g/kg, 60 min before test	1 day	7 (male)	19.7 ± 1.7	Wrestling	National	Wingate test	 Anaerobic power  Body mass  Urine specific gravity  Saliva osmolality
Mixed combat sports
[75] de Azevedo et al. (2019)	Caffeine	5 mg/kg, 60 min before test	1 day	11 (male)	27.6 ± 4.3	MMA	Professional	Punching exercise protocol	 Punch frequency  Mean and maximum punching force  Readiness to invest effort  Perceptual training intensity
[76] Chycki et al. (2020)	Sodium Bicarbonate	10 g (2 × 5 g), 90 min before test	21 days	16 (male)	24.3 ± 0.5	Combat sports	International	Wingate test + Cognitive performance test	↑ Total work in upper limb↑ Peak and Mean Power in upper limb  Total work in lower limb  Peak and Mean Power in lowerlimb↑ Blood lactate ↑ IGF-1 ↓ Cortisol↓ BDNF  Display time in cognitive tests
[77] de Oliveira et al. (2020)	Beetroot-based gel	12.2 ± 0.2 mmol Nitrate, 120 min before test	1 day	14 (male)	29.9 ± 8.5	Combat sports	Amateur	Forearm muscle isometric strength test + Handgrip isotonic exercise	↑ Maximal voluntary forearm contraction  Muscle O_2_ saturation  Blood volume in forearm  Time until fatigue

**Table 3 nutrients-14-02588-t003:** PEDro scale scores by items.

Study	Criteria											TOTAL
	1	2	3	4	5	6	7	8	9	10	11	
[23] Ouergui et al., 2022	no	yes	n/a	yes	yes	yes	n/a	yes	yes	yes	yes	8
[24] Jodra et al., 2020	yes	yes	n/a	yes	yes	yes	n/a	yes	yes	yes	yes	8
[25] Park et al., 2020	no	no	no	yes	yes	no	no	n/a	yes	yes	no	4
[26] San Juan et al., 2019	no	yes	n/a	yes	yes	yes	n/a	yes	yes	yes	yes	8
[27] Rezaei et al.,2019	yes	yes	n/a	yes	yes	yes	n/a	yes	yes	yes	yes	8
[28] Coswig et al., 2018	no	yes	n/a	yes	yes	yes	n/a	yes	yes	yes	yes	8
[29] Lopes-Silva et al., 2015	no	yes	n/a	yes	yes	yes	n/a	yes	yes	yes	yes	8
[30] Santos et al., 2014	no	yes	yes	yes	yes	yes	n/a	yes	yes	yes	yes	9
[31] Sarshin et al., 2021	yes	yes	no	yes	no	no	no	yes	yes	yes	yes	7
[32] Gough et al., 2019	no	no	n/a	yes	yes	yes	n/a	yes	yes	yes	yes	7
[33] Lopes-Silva et al., 2018	no	yes	n/a	yes	yes	yes	n/a	yes	yes	yes	yes	8
[34] Siegler et al., 2010	no	yes	n/a	yes	yes	yes	n/a	yes	yes	yes	yes	8
[35] Alabsi et al., 2022	yes	yes	n/a	yes	yes	yes	n/a	yes	yes	yes	yes	9
[36] Kim et al., 2018	no	no	n/a	yes	yes	yes	n/a	yes	yes	yes	yes	7
[37] Donovan et al., 2012	no	yes	n/a	yes	yes	no	no	yes	yes	yes	yes	7
[38] Manjarrez-Montes de Oca et al., 2013	yes	yes	n/a	yes	yes	yes	n/a	no	yes	yes	yes	7
[39] Miraftabi et al., 2021	yes	yes	n/a	yes	yes	yes	n/a	no	yes	yes	yes	8
[40] Antonietto et al., 2021	yes	yes	n/a	yes	yes	yes	n/a	yes	yes	yes	yes	9
[41] Tatlici et al.,2019	yes	yes	n/a	yes	yes	no	no	yes	yes	yes	yes	7
[42] Merino Fernández et al., 2022	no	yes	n/a	yes	yes	yes	yes	yes	yes	yes	yes	9
[43] Krawczyk et al., 2022	yes	yes	n/a	yes	yes	yes	n/a	yes	yes	yes	yes	9
[44] Merino Fernandez et al., 2021	yes	n/a	n/a	yes	yes	yes	n/a	yes	yes	yes	yes	8
[45] Lopes-Silva et al., 2021	no	no	n/a	yes	yes	yes	n/a	yes	yes	yes	yes	7
[46] Filip-Stachnik et al., 2021	yes	yes	n/a	yes	yes	yes	n/a	yes	yes	yes	yes	9
[47] Carmo et al., 2021	yes	yes	n/a	yes	yes	yes	n/a	yes	yes	yes	yes	9
[48] Negaresh et al., 2018	yes	yes	n/a	yes	yes	yes	n/a	yes	yes	yes	yes	8
[49] Durkalec-Michalski et al., 2019	yes	yes	n/a	yes	yes	yes	n/a	no	no	yes	yes	6
[50] Saldanha da Silva et al., 2019	no	yes	n/a	yes	yes	yes	n/a	yes	yes	yes	yes	8
[51] Saldanha da Silva et al., 2018	no	yes	n/a	yes	yes	yes	n/a	yes	yes	yes	yes	8
[52] Astley et al., 2017	no	yes	n/a	yes	yes	yes	n/a	yes	yes	yes	yes	8
[53] Diaz-Lara et al., 2016	no	yes	n/a	yes	yes	yes	yes	yes	yes	yes	yes	9
[54] Felippe et al.,2016	no	yes	n/a	yes	yes	yes	n/a	yes	yes	yes	yes	8
[55] Diaz-Lara et al., 2015	no	yes	yes	yes	yes	yes	yes	yes	yes	yes	yes	10
[56] Lopes-Silva et al., 2014	no	no	n/a	yes	yes	yes	n/a	no	yes	yes	yes	6
[57] Aedma et al.,2013	no	yes	n/a	yes	yes	yes	n/a	yes	yes	yes	yes	8
[58] Souissi et al., 2012	yes	n/a	n/a	yes	no	no	no	yes	yes	yes	yes	5
[59] Ragone et al., 2020	no	yes	n/a	yes	yes	yes	n/a	yes	yes	yes	yes	8
[60] Durkalec-Michalski et al., 2020	yes	yes	n/a	yes	yes	yes	n/a	yes	yes	yes	yes	8
[61] Durkalec-Michalski et al., 2018	yes	yes	n/a	yes	yes	yes	n/a	no	yes	yes	yes	7
[62] Tobias et al., 2013	yes	yes	n/a	yes	yes	yes	n/a	yes	yes	yes	yes	8
[63] Artioli et al., 2007	no	no	n/a	yes	yes	yes	n/a	yes	yes	yes	yes	7
[64] Aedma et al., 2014	no	yes	n/a	yes	yes	yes	n/a	yes	yes	yes	yes	8
[65] Timpmann et al., 2012	no	no	n/a	yes	yes	yes	n/a	yes	yes	yes	yes	7
[66] Andrade Kratz et al., 2016	no	yes	yes	yes	yes	yes	n/a	yes	yes	yes	yes	9
[67] Kern et al., 2011	no	yes	n/a	yes	yes	yes	n/a	yes	yes	yes	yes	8
[68] Aedma et al., 2015	no	yes	n/a	yes	yes	yes	n/a	yes	yes	yes	yes	8
[69] Abbasalipuor et al., 2012	no	yes	n/a	no	no	no	no	yes	yes	yes	yes	5
[70] Tatlici et al., 2021	yes	yes	n/a	yes	yes	yes	n/a	yes	yes	yes	yes	9
[71] de Oliveira et al., 2018	yes	yes	n/a	yes	yes	yes	n/a	yes	yes	yes	yes	8
[72] Yavuz et al.,2014	no	yes	n/a	n/a	n/a	no	no	yes	yes	yes	yes	5
[73] Liu et al., 2009	no	yes	n/a	yes	no	no	no	yes	yes	yes	yes	6
[74] McKenna et al., 2017	no	yes	n/a	yes	no	no	no	yes	yes	yes	yes	6
[75] Azevedo et al., 2019	no	no	n/a	yes	yes	yes	n/a	yes	yes	yes	yes	7
[76] Chycki et al., 2020	yes	yes	n/a	yes	yes	no	no	yes	yes	yes	yes	8
[77] Vieira de Oliveira et al., 2020	yes	yes	n/a	yes	yes	yes	n/a	yes	yes	yes	yes	8

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
