# Peer review of "Nutritional Ergogenic Aids in Combat Sports: A Systematic Review and Meta-Analysis"

_nutrients, 2022, doi:10.3390/nu14132588_

Round 1

Reviewer 1 Report

Line 39 – remove double spacing

Line 45 – rather use require than show

Line 66 – remove one judo

Line 68 – no need for the before legs… This manuscript needs to be looked at by a native English speaker

Line 70 – again double spacing… please check for it throughout this paper

Line 75 – you need to be more specific here. For instance, weight categories in combat sports are established in order to match combat sports athletes who similar physical characteristics and thereby emphasize fair play… Please rephrase accordingly

Line 77,78 – this is an important point and you need to elaborate further on this in the discussion section. Is weight cycling some form of ergogenic aid as well? There are many publications on this topic (10.1007/s40279-016-0541-x., 10.1186/s13102-021-00381-2., 10.1123/ijsnem.2018-0165.)

Line 84 – whose, not those

Line 269 – you mean as for?

Line 412 – sentence unclear. Rephrase it.

Line 435 – citation needed here

Line 464 – rephrase

Line 484 – rephrase

Line 490 – rephrase

Line 525 – rephrase

Line 563 – overall, this is a really good study but oftentimes the wording is not the best. Therefore, proofreading by English speaking professional would help this paper reach its full potential. Also, a paragraph about tolerance developed by frequent caffeine users should be discussed and how this occurrence could decrease the possible benefits of caffeine intake. Nevertheless, a few more sentences about the mechanism of caffeine action on performance should be discussed, especially in combat sports.

Importantly, references are not correctly done! Please revise the entire reference section!

Author Response

We thank the reviewer for her/his efforts in revising our paper in a detailed and careful form. Thank you so much.

Line 39 – remove double spacing

Following the comment from the reviewer, it has been modified.

Line 45 – rather use require than show

Following the comment from the reviewer, it has been modified for a better comprehension.

Line 66 – remove one judo

Following the comment from the reviewer, it has been modified.

Line 68 – no need for the before legs… This manuscript needs to be looked at by a native English speaker.

Following the comment from the reviewer, paper has been proofreader by English native.

Line 70 – again double spacing… please check for it throughout this paper

Following the comment from the reviewer, it has been modified.

Line 75 – you need to be more specific here. For instance, weight categories in combat sports are established in order to match combat sports athletes who similar physical characteristics and thereby emphasize fair play… Please rephrase accordingly

Thank you for your comment, the paragraph has been modified.

Line 77,78 – this is an important point and you need to elaborate further on this in the discussion section. Is weight cycling some form of ergogenic aid as well? There are many publications on this topic (10.1007/s40279-016-0541-x., 10.1186/s13102-021-00381-2., 10.1123/ijsnem.2018-0165.)

We thank the reviewer for these interesting papers. The authors believe that the main objective of the study is exclusively type A ergo-nutritional aids according to the AIS, and these do not refer to effects on weight control in athletes. That is why we think that discussing non-nutritional ergogenic aids for weight control in combat athletes is outside the main objective of this study and would increase its extension, being already quite high.

Line 84 – whose, not those

Following the comment from the reviewer, it has been modified for a better comprehension.

Line 269 – you mean as for?

Following the comment from the reviewer, it has been modified for a better comprehension.

Line 412 – sentence unclear. Rephrase it.

Following the comment from the reviewer, it has been modified for a better comprehension.

Line 435 – citation needed here

Following the comment from the reviewer, reference has been added for a better comprehension (Hadzic, 2019).

Line 464 – rephrase

Following the comment from the reviewer, it has been modified for a better comprehension.

Line 484 – rephrase

Following the comment from the reviewer, it has been modified for a better comprehension.

Line 490 – rephrase

Following the comment from the reviewer, it has been modified for a better comprehension.

Line 525 – rephrase

Following the comment from the reviewer, it has been modified for a better comprehension.

Line 563 – overall, this is a really good study but oftentimes the wording is not the best. Therefore, proofreading by English speaking professional would help this paper reach its full potential.

Following the comment from the reviewer, paper has been proofreader by English native.

Also, a paragraph about tolerance developed by frequent caffeine users should be discussed and how this occurrence could decrease the possible benefits of caffeine intake. Nevertheless, a few more sentences about the mechanism of caffeine action on performance should be discussed, especially in combat sports.

We thank the reviewer for this comment. Following the reviewer’s suggestion, in the discussion section we have added some lines (L 446) about caffeine consumption by coffee consumers to clarify this aspect.

On the other hand, we have added more information about the relation of caffeine action on performance using a recent published study (L 470).

Importantly, references are not correctly done! Please revise the entire reference section!

Following the reviewer’s indication, the reference section was revised and formatted.

Reviewer 2 Report

The manuscript needs extensive revision. Most of the writing is general (colloquial in places).

Please follow author guidelines for the abstract. Headings are not required.

Search was completed March 2021. I suggest to update.

It needs to be clarified whether participants in the studies were biased based on adverse symptoms, e.g. in the case of sodium bicarbonate, known to give in some participants gastrointestinal symptoms.

L25. Please move “(5 - 10 mg/kg).” after mention of caffeine. Please clarify whether that is acute or chronic intake of caffeine.

L26. Please be specific which buffering supplements. Not the effect of these supplements is buffering so I suggest to rephrase.

L27. What are “grappling disciplines”. Please clarify.

L28. “Other NEAs need further study”. This is too general. Please revise.

Please clarify the abstract what performance aspects are important to enhance by supplements on combat sports.

Please clarify what particular performance is enhanced in combat sports with intake of caffeine.

Please justify why this review is needed. Note doi: 10.1080/10408398.2022.2068499m, a systematic review and meta-analysis on caffeine in combat sports. In addition, the introduction needs to be more focussed on what parameters are specifically of interest. In Table 2 for example, it seems that all findings from studies are reported.

L36. Ref 1 is not on the different rules for combat sports. Note it is not the task for the reviewer to check every reference but not good to note that use of Ref 1 does not seem to be the original source to support the provided statement.

L48. Ref 2 has no original data on lactate. Please clarify when review papers are being used throughout the manuscript.

L71. “both metabolisms”. Please clarify.

L81. To describe an ergogenic aid as a training method is problematic. You mean sprint interval training would be an ergogenic aid. Please revise.

L85. Please elaborate what harmful effects are considered by athletes to avoid with the intake of supplements?

L123. What were the specific between differences? Please provide at least an example.

L132. Studies that have participants not taking any supplements as the control condition are problematic when the aim is to examine performance. I suggest to clarify which studies had that approach but better would be to remove those from the review.

L138. Are you sure “gray literature” is appropriate phrase for still academic output. Please check.

L189. Change to “electromyography”

Table 2. Change “varaibility “ to “variability”

Table 2. Do we need the information on urine parameters. The introduction does not justify the interest in such parameters.

But in L132 is stated “not including any ergogenic aids classified within group A by the AIS because of their high evidence grade”. As far as I know, arginine is not on the list. Please clarify.

L223. I suggest to provide quantitative information, e.g. “more number of attacks”. What was the percentual changes and maybe some comment on whether such changes are meaningful for the sport. Revise this throughout the manuscript as otherwise all the information (e.g. L226, “were higher than control group”), it is difficult to grasp the importance of the change.

L250 “been used in two assays”. Is this correct?

L260. Change to “did not offer”.

L379. Is “lactate acid” the same as “lactate”? For lactate measured in blood samples, it needs to be clarified whether it was blood lactate or plasma lactate. Check whether studies used lysing agents. This needs substantial clarification.

L466. I suggest not to refer to observations in non-combat sports because the information provided is already challenging enough for the “non-combat” reader.

L510. Please elaborate on “more quality in the experimental design”. What were the issues?

L538. “these disciplines”. Please be specific.

L583. There is inconsistency in the references, e.g. abbreviations and full names of journals. References should only have one number.

Author Response

We thank the reviewer for her/his efforts in revising our paper in a detailed and careful form. Thank you so much.

The manuscript needs extensive revision. Most of the writing is general (colloquial in places).

Following the comment from the reviewer, paper has been proofreader by English native.

Please follow author guidelines for the abstract. Headings are not required.

Following the comment from the reviewer, it has been modified.

Search was completed March 2021. I suggest to update.

Thank you very much for your advice. We decided to set a deadline so as not to have to continuously modify the article, but after a delay beyond our control, we believe that after a year it is a good idea to do the revision again. Following the comment from the reviewer, the systematic review has been done again obtaining 14 news articles that they have been added to the manuscript. The appropriate modifications have been made in all the sections.

It needs to be clarified whether participants in the studies were biased based on adverse symptoms, e.g. in the case of sodium bicarbonate, known to give in some participants gastrointestinal symptoms.

The authors appreciate the reviewer´s comment. Most of them did not report any gastrointestinal symptom. In any case, we have been incorporated as a supplementary file a table regarding this issue.

L25. Please move “(5 - 10 mg/kg).” after mention of caffeine. Please clarify whether that is acute or chronic intake of caffeine.

Following the comment from the reviewer, it has been modified for a better comprehension.

L26. Please be specific which buffering supplements. Not the effect of these supplements is buffering so I suggest to rephrase.

Following the comment from the reviewer, supplements have been specified.

In our opinion, the term "buffering supplements" is valid, since their final function is to buffer the excess of H+ coming from glycolytic metabolism. Both sodium bicarbonate, sodium citrate and beta-alanine are often indicated as "buffering supplements" in several studies, due to their role in buffering pH in high-intensity exercises:

https://pubmed.ncbi.nlm.nih.gov/34687438/

https://doi.org/10.3389/fnut.2021.669102

L27. What are “grappling disciplines”. Please clarify.

Following the comment from the reviewer, “grappling disciplines” have been changed by “grappling combat sports” which is a term used in the literature:

https://pubmed.ncbi.nlm.nih.gov/35222096/

L28. “Other NEAs need further study”. This is too general. Please revise.

Following the comment from the reviewer, we have listed ergogenic aids specifically

Please clarify the abstract what performance aspects are important to enhance by supplements on combat sports.

Following the comment from the reviewer, we have listed important performance aspects that could be improved with NEAs.

Please clarify what particular performance is enhanced in combat sports with intake of caffeine.

Following the recommendation from the reviewer, we have indicated in the abstract the particular performance aspects that were improved with the use of caffeine in the dosage indicated and in regard of the new meta-analysis results.

Please justify why this review is needed. Note doi: 10.1080/10408398.2022.2068499m, a systematic review and meta-analysis on caffeine in combat sports. In addition, the introduction needs to be more focussed on what parameters are specifically of interest. In Table 2 for example, it seems that all findings from studies are reported.

We want to thanks the comment of the review. This specific meta-analysis of caffeine in combat sports was published after the initial deadline of our first systematic review, generally with ergo-nutritional aids classified as type A by the AIS, and we did not have the opportunity to contrast and discuss our results with this recent meta-analysis. After the great advice of the reviewer to extend the time frame for the systematic search to the present, we have found fourteen new studies, of which seven pertain to caffeine. For this reason, we have carried out a new meta-analysis, focusing on lactate levels as a common parameter in most of the studies found and a marker of the state of glycolytic metabolism. In addition, our meta-analysis, being more recent, has a greater number of studies, so it satisfactorily completes the article indicated by the reviewer. The articles not used in the meta-analysis by Díaz-Lara (2022) are listed below because they are more recent and that we have incorporated into our study:

- Sandanha da Silva 2019: https://doi.org/10.1177/0031512519826726

- San Juan 2019: https://doi.org/10.3390/nu11092120

- Lopes-Silva 2021: https://doi.org/10.1080/17461391.2021.1874058

- Rezaei 2019: https://doi.org/10.1186/s12970-019-0313-8

Table 2 collects, not only the results on the use of caffeine as NEA, but also that of other supplements with a high degree of evidence on its efficacy. The authors think that the exhaustive exposition of all the exposed results greatly enriches the work, as well as its quality.

L36. Ref 1 is not on the different rules for combat sports. Note it is not the task for the reviewer to check every reference but not good to note that use of Ref 1 does not seem to be the original source to support the provided statement.

 Thank you very much for your comment. In this reference put literally the follow:

“Combat sports include combinations of movement by the whole body and limbs (upper and lower). The pattern of effort is characterized by multiple high-intensity episodes of short duration (rounds) interspersed by intervals that, depending on the specific combat sport and specific regulations, can range from seconds to minutes.” 

Therefore, we believe that the reference is correct in this paragraph.

L48. Ref 2 has no original data on lactate. Please clarify when review papers are being used throughout the manuscript.

 Thank you very much for your comment. In this reference put literally the follow:

“Additionally, the high-intensity component of combat sport competitions induces significant anaerobic strain with research observing high levels of blood lactate (>12 mmol.L-1) following competition (Bouhlel et al., 2006Hanon et al., 2015).” 

This reference (Barley, 2019) encompasses the other two and is much more current, following the advice of the journal in terms of indicating the most up-to-date references.

L71. “both metabolisms”. Please clarify.

Following the comment from the reviewer, it has been modified for a better comprehension.

L81. To describe an ergogenic aid as a training method is problematic. You mean sprint interval training would be an ergogenic aid. Please revise.

Following the advice from the reviewer, it has been modified for a better comprehension.

L85. Please elaborate what harmful effects are considered by athletes to avoid with the intake of supplements?

Following the advice from the reviewer, it has been listed harmful effects that supplements try to avoid during the exercise.

L123. What were the specific between differences? Please provide at least an example.

Following the advice from the reviewer, it has been exposed one argument about the discrepancies raised during the systematic search. For example, when a study shows the effect of a multi ingredient supplement, where the risk of masking the real effect of a NEA could be affected.

L132. Studies that have participants not taking any supplements as the control condition are problematic when the aim is to examine performance. I suggest to clarify which studies had that approach but better would be to remove those from the review.

Following the advice from the reviewer, this issue has been verified, and the data reported are from PLACEBO group.

L138. Are you sure “gray literature” is appropriate phrase for still academic output. Please check.

We appreciate the comment of the reviewer and we understand the doubts that arise with this term. In any case, this term is stablished in the research literature:

 https://pubmed.ncbi.nlm.nih.gov/28857505/

Moreover, this term is used in the methodology description of many review studies published in this journal:

https://pubmed.ncbi.nlm.nih.gov/31242624/

L189. Change to “electromyography”

Following the comment from the reviewer, it has been modified for a better comprehension.

Table 2. Change “varaibility “ to “variability”

Following the comment from the reviewer, it has been modified for a better comprehension. 

Table 2. Do we need the information on urine parameters. The introduction does not justify the interest in such parameters.

Thank you for your comment. Although the evaluation of urine parameters is not a main objective, we think that, incidentally, this study can provide some information on whether the use of certain NEAs at certain doses can affect the water homeostasis of the athlete, as later We indicate in the discussion. For example, caffeine has had certain suspicions that it could affect the water status of the athlete:

https://pubmed.ncbi.nlm.nih.gov/25154702/

But in L132 is stated “not including any ergogenic aids classified within group A by the AIS because of their high evidence grade”. As far as I know, arginine is not on the list. Please clarify.

In the discussion (Lines 603-606) the reason for the inclusion of arginine in the review is indicated, since it belongs to the substances that enhance NO synthesis. It is not really included in group A, but it deserves a special mention as it is included in this type of substance and as there are currently several publications with contradictory results, so this review could shed some light on combat sports:

https://pubmed.ncbi.nlm.nih.gov/35308069/

https://pubmed.ncbi.nlm.nih.gov/33815657/

https://pubmed.ncbi.nlm.nih.gov/35337088/

https://pubmed.ncbi.nlm.nih.gov/35457330/

L223. I suggest to provide quantitative information, e.g. “more number of attacks”. What was the percentual changes and maybe some comment on whether such changes are meaningful for the sport. Revise this throughout the manuscript as otherwise all the information (e.g. L226, “were higher than control group”), it is difficult to grasp the importance of the change.

We appreciate the reviewer´s suggestion and we have added information of % of changes and numeric values throughout the results section.

L250 “been used in two assays”. Is this correct?

Following the comment from the reviewer, it has been modified for a better comprehension. 

L260. Change to “did not offer”.

Following the comment from the reviewer, it has been modified for a better comprehension. 

L379. Is “lactate acid” the same as “lactate”? For lactate measured in blood samples, it needs to be clarified whether it was blood lactate or plasma lactate. Check whether studies used lysing agents. This needs substantial clarification.

The authors thank the reviewer´s indication and we have reviewed this issue. We have homogenized the terms about lactate in the review, following the same considerations that the paper recommended by the reviewer:  doi: 10.1080/10408398.2022.2068499m

L466. I suggest not to refer to observations in non-combat sports because the information provided is already challenging enough for the “non-combat” reader.

We understand the reviewer's suggestion and appreciate it. The authors believe that it is necessary to make comparisons with other sets of sports disciplines, which due to their characteristics, can resemble those of combat, and assess whether there is a parallelism between the results obtained. This enriches the discussion and gives a global perspective of the use of NEAs at a general level.

L510. Please elaborate on “more quality in the experimental design”. What were the issues?

Thank you for your comment. Following the comment from the reviewer we have detailed the sentence mentioned above.

L538. “these disciplines”. Please be specific.

Following the comment from the reviewer, it has been modified for a better comprehension. 

L583. There is inconsistency in the references, e.g. abbreviations and full names of journals. References should only have one number.

Following the reviewer’s indication, the reference section was revised and formatted.

Round 2

Reviewer 2 Report

Thanks for responding to my comments and suggestions. However, there is some clarity needed.

L140. Studies that have participants not taking any supplements as the control condition are problematic when the aim is to examine performance. I suggest to clarify which studies had that approach but better would be to remove those from the review. If you decide not to remove them, then make clear which studies did have as control not taking supplementation. Studies in which participants were not supplemented cannot be referred to as placebo. 

You did respond with Following the advice from the reviewer, this issue has been verified, and the data reported are from PLACEBO group.”

But there is still mention in the inclusion L140 “.. or not receiving supplementation”

Please clarify whether only placebo controlled studies were used.

L183. “41 met all inclusion criteria”. However, Figure 1 indicates 55. Please ensure numbers in the text match with numbers in Figure 1.

Table 2. Change “mg/kg/” to “mg/kg” throughout the Table or to “mg/kg,” as time information follows.

L393. Change “checking” to “examining”.

L432. Change “lactic acid “to “lactate”.

References should only have one number.

Author Response

We thank the reviewer for her/his additional advices to improve the manuscript. Thank you so much. 

L140. Studies that have participants not taking any supplements as the control condition are problematic when the aim is to examine performance. I suggest to clarify which studies had that approach but better would be to remove those from the review. If you decide not to remove them, then make clear which studies did have as control not taking supplementation. Studies in which participants were not supplemented cannot be referred to as placebo. 

You did respond with Following the advice from the reviewer, this issue has been verified, and the data reported are from PLACEBO group.”

But there is still mention in the inclusion L140 “.. or not receiving supplementation”

Please clarify whether only placebo controlled studies were used.

The authors understand the clarification that the reviewer demands from us. For this reason, we have attached a table indicating all the selected studies and whether they use a control or placebo group. Most of the selected articles used exclusively a placebo group. On the other hand, there are studies that have a control group and a placebo group, but all the results offered in Table 2 refer to significant changes compared to the placebo group. Finally, only one study (Abbasalipuor, 2021) has used only a control group, which is indicated in the L389, and the authors have also specified its fair PEDRO scale (L566). To conclude, the authors have replaced the expression "control group" with "placebo group" throughout the manuscript.

L183. “41 met all inclusion criteria”. However, Figure 1 indicates 55. Please ensure numbers in the text match with numbers in Figure 1.

Following the comment from the reviewer, we have changed the numbers of studies selected correctly (from L183 to L189)

Table 2. Change “mg/kg/” to “mg/kg” throughout the Table or to “mg/kg,” as time information follows.

Following the comment from the reviewer, it has been modified.

L393. Change “checking” to “examining”.

Following the comment from the reviewer, it has been modified for a better comprehension.

L432. Change “lactic acid “to “lactate”.

Following the comment from the reviewer, it has been modified for a better comprehension.

References should only have one number.

Following the comment from the reviewer, it has been modified.

Round 3

Reviewer 2 Report

Thanks for addressing all my comments and suggestions.